# Covered Rutile-TiO_2_ Nanoparticles Enhance Tomato Yield and Growth by Modulating Gas Exchange and Nutrient Status

**DOI:** 10.3390/plants12173099

**Published:** 2023-08-29

**Authors:** Eneida A. Pérez-Velasco, Luis A. Valdez-Aguilar, Rebeca Betancourt-Galindo, José Antonio González-Fuentes, Adolfo Baylón-Palomino

**Affiliations:** 1Departamento de Materiales Avanzados, Centro de Investigación en Química Aplicada, Saltillo 25294, Mexico; 2Departamento de Horticultura, Universidad Autónoma Agraria Antonio Narro, Saltillo 25315, Mexico; jagf252001@gmail.com; 3Departamento de Biociencias y Agrotecnia, Centro de Investigación en Química Aplicada, Saltillo 25294, Mexico; adolfo.baylon@ciqa.edu.mx

**Keywords:** surface covering, maltodextrin, nutrient concentration, SPAD, photosynthesis, titanium dioxide nanoparticles

## Abstract

Nanotechnology has developed materials that can increase food production while reducing the use of conventional fertilizers. In this study, the effect of two forms of application (foliar and drench) as well as covering or non-covering of the surface of titanium dioxide nanoparticles (nTiO_2_) with maltodextrin (MDX) at 1500 ppm was investigated on tomato plants. The results show that treatment of tomato with nTiO_2_ increased yield (+21%), while covering the surface of the NPs resulted in a further yield increase (+27%). Similar trends were observed in the dry weight of vegetative plant parts. Fruit firmness (+33%) and total soluble solids (+36%) were enhanced by MDX-covered nTiO_2_. Application of nTiO_2_ resulted in enhanced SPAD index, photosynthesis rate, NO_3_^−^, K, and Ca concentration in the petiole sap, whereas in the fruits there was an increase in P and K in MDX-covered nTiO_2_. Considering the dilution effect due to the higher fruit yield, N, P, Mg, Cu, and B increased in plants treated with nTiO_2_. Covering the surface with MDX resulted in an enhanced response to nTiO_2_, as fruit yield and quality increased compared to plants treated with non-covered nTiO_2_.

## 1. Introduction

The challenges of chemical pollution, climate change, natural resources abuse, and urbanization pose significant threats to the food production required to satisfy the needs of a population projected to reach 9.7 billion individuals by the year 2050 [1,2]. Agricultural research is now engaged in the development of novel technologies aimed at enhancing food production while concurrently mitigating environmental degradation [3]. Nanotechnology (NT) is an emergent technology that seeks to enhance conventional food production by the development of materials at nanoscale sizes ranging from 1 to 100 nm, which possess distinctive physical and chemical properties [4,5].

In the agricultural sector, nanoparticles (NPs) are utilized as nanosensors, nanopesticides, and nanofertilizers [6]. Nanosensors provide the potential to address several agricultural challenges, such as nutrient shortages or toxicities, early identification of plant diseases and pests, and real-time monitoring of extreme weather conditions [2,7]. By using nanosensors, producers may effectively safeguard their crops by promptly detecting and responding to these issues [2,7]. Nanopesticides have the potential to serve as active components of pesticides, and differ from traditional pesticides because they have enhanced dispersion, wettability, stability, permeability, and biodegradability [3,8]. Nanofertilizers differ from conventional fertilizers in that they demonstrate higher absorption and reduced leaching, thereby increasing crop quality and yield, and may include one or more essential/beneficial nutrients for plant growth and development [9].

Nanoparticles of titanium dioxide (nTiO_2_) and silicon dioxide (nSiO_2_) are widely used and are produced in large volumes [10]; around 88,000 tons of nTiO_2_ are annually employed in various applications, including the manufacturing of solar cells, as additives in the food sector (e.g., dyes), utilization in paints, and incorporation in personal care items such as sunscreens and disinfectants [11]. Titanium (Ti) is not classified as an essential nutrient; however, there are documented reports of its favorable impacts on cultivated plants. In spinach (*Spinacia oleracea* L.), nTiO_2_ resulted in a 4% increase in seed germination, a 76% increase in dry weight, and a 44.5% increase in chlorophyll concentration [12]. Likewise, in *Lactuca sativa* L. the application of nTiO_2_ led to a 49% increase in shoot length, a 62% increase in root length, and a 175% increase in phosphorus (P) content [13]. Barley (*Hordeum vulgare* L.) seed treated with nTiO_2_ showed a notable increase in the levels of glutamic acid (+33%), and glycine (+29%), as well as the concentrations of nitrogen (N) (+25%), calcium (Ca) (+85%), iron (Fe) (+424%), and zinc (Zn) (+25%) [14]. The application of nTiO_2_ has been shown to improve physiological parameters in several plant species. For instance, in tomato (*Solanum lycopersicum* L.) the photosynthesis rate was seen to increase by 18.2% [15]. Similarly, in mint (*Mentha piperita* L.) [16], radish (*Raphanus sativus* L.) [17], and spinach [12] the photosynthesis rate showed an increase of 23.8%, 50%, and 313%, respectively. In maize (*Zea mays* L.), the contents of carotenoids and chlorophyll exhibited respective increases of 37% and 5.3% [18].

The clear impacts of nTiO_2_ on plant growth and physiology are indicative of its biostimulant properties. However, there is a major problem throughout the process of synthesis of the nanoparticles of TiO_2_ because the particles have a tendency to agglomerate, leading to a decrease in their dispersibility and stability. According to Kango et al. [19], NPs exhibit a pronounced inclination towards agglomeration, resulting in insufficient dispersion within the polymer matrix that results in the deterioration of both the optical and mechanical characteristics of the nanocomposites. To enhance the stability of nanoparticle dispersion in aqueous media or polymer matrices it is crucial to employ surface modification techniques involving polymer surfactant molecules or other modifiers in order to induce a strong repulsive force between nanoparticles, thereby mitigating agglomeration [19]. To mitigate the agglomeration and to enhance the stability of nTiO_2_, the surface can be coated with modifying agents, which can be either inorganic (such as SiO_2_ and Al_2_O_3_) or organic (such chitosan and starch) [20]. The present study was designed to assess the effect of maltodextrin (MDX), a polysaccharide derived from maize starch, in forming a coating on the surface of the nanoparticles in order to reduce agglomeration and thereby improve the dispersibility and stability of nTiO_2_. Additionally, this study aimed to examine the effects of different application methods, specifically foliar and drench treatments of nTiO_2_ on tomato plants, with the objective of evaluating the effects on plant growth, nutritional status, gas exchange parameters, and fruit production.

## 2. Results

### 2.1. Characterization of nTiO_2_

The TiO_2_ nanoparticles exhibited a rod-like shape with an average length of 76 nm ranging from 10 to 190 nm and an average width of 8.52 nm ranging from 1 to 20 nm (Figure 1). The crystalline structure of nTiO_2_ was determined through XRD pattern analysis. The analysis revealed diffraction peaks that matched the rutile TiO_2_ structure (Figure 2), which aligns with the parameters established by the Joint Committee on Powder Diffraction Standards (JCPDS 01-077-0443) [21,22]. The surface coating of MDX did not have an impact on the crystalline structure of nTiO_2_. (Figure 2).

The Fourier transform infrared analysis revealed the presence of the –OH stretching vibration band at approximately 3430 cm^−1^, which is a characteristic of nTiO_2_ (Figure 3) [23,24]. However, in the case of MDX-surface covered nTiO_2_ two bands were observed: the stretching vibration at around 3430 cm^−1^, and an additional band at 1000 cm^−1^, which can be attributed to MDX (Figure 3). These findings provide evidence that the NPs were effectively coated with MDX, as demonstrated by the presence of absorption bands corresponding to both nTiO_2_ and MDX. The spectral analysis additionally indicated that the introduction of MDX for surface modification did not have any impact on the crystalline structure of the NPs, as shown in Figure 3.

### 2.2. Effects of nTiO_2_ on Fruit Yield and Quality

The application of nTiO_2_ by both leaf and drench methods resulted in an increase in fruit yield and individual fruit weight (Table 1). The orthogonal contrasts indicate that the yield might be further enhanced through covering the surface of nTiO_2_ with MDX, particularly when administered via drenching (Table 1). Additionally, the application of either MDX-covered or non-covered nTiO_2_, as well as both leaf and drench applications, resulted in an increase in fruit size as measured by polar and equatorial diameters. In contrast, the application of MDX-covered nTiO_2_ by drenching resulted in the production of fruits with larger diameters compared to plants treated with non-surface covered nTiO_2_. The application of non-MDX-covered nTiO_2_ resulted in an enhancement in fruit quality, as indicated by an increase in firmness and total soluble solids; however, the use of MDX-covered nTiO_2_ led to an even greater improvement in firmness and total soluble solids content. The impact on the total concentration of soluble solids was shown to be independent of the method of application, whether by foliar spraying or drenching, when MDX-coated nTiO_2_ was utilized (Table 1).

### 2.3. Effects of nTiO_2_ on Growth and Biomass

The application of nTiO_2_ on the leaves resulted in a significant increase in plant height and stem diameter of 11.4% and 12.3% respectively, compared to plants that did not receive this treatment (Table 2). The drench application of surface-covered nTiO_2_ significantly increased other plant parameters, including the dry weight of leaves, stems, and roots, as well as the total biomass and SPAD index (Table 2). Both stem diameter and SPAD index exhibited an increase in the presence of nTiO_2_ regardless of whether it was surface or non-surface covered (Table 2). The plants treated with nTiO_2_ exhibited an increase in SPAD index irrespective of the method of application or surface coverage (Table 2). These results indicate that the application of MDX-covered nTiO_2_ through drench resulted in the highest fruit yield and weight compared to control plants (Table 1), as well as the highest total, stem, leaf, and root dry weight of plants (Table 2); however, orthogonal contrast analysis showed that the use of MDX-covered nTiO_2_ did not significantly enhance plant growth parameters except for stem diameter and root dry weight compared to the application of non-covered nTiO_2_ (Table 2).

### 2.4. Effect of NPs-TiO_2_ on Gas Exchange Parameters

The photosynthetic rate exhibited a significant increase in all experimental treatments that included the application of nTiO_2_ irrespective of the presence of MDX covering and the method of application (Table 3). The addition of nTiO_2_ demonstrated a positive impact on the photosynthetic process, as evidenced by its correlation with the SPAD index (Figure 4A), yield (Figure 4B,C), internal CO_2_ concentration (Figure 4D), and biomass accumulation (Table 2). The application of MDX on the surface of nTiO_2_ did not result in any additional enhancement of the photosynthetic rate (Table 3). However, both the stomatal conductance and transpiration rate exhibited considerable increases when either MDX-covered or non MDX-covered nTiO_2_ was applied (Table 3). The application of MDX-covered nTiO_2_ resulted in a greater increase compared to non MDX-covered nTiO_2_, particularly when delivered through drenching (Table 3). In general, the internal CO_2_ concentration exhibited a reduction when foliar application of surface MDX-covered nTiO_2_ was employed as well as when non-covered nTiO_2_ was administered by drenching (Table 3). The results from orthogonal contrasts further support the finding that the internal CO_2_ concentration declined irrespective of whether covered or non-covered nTiO_2_ was used for application (Table 3).

### 2.5. Effect of nTiO_2_ on Nutrient Status of Petiole Sap

According to the orthogonal contrasts, the concentrations of nitrate (NO_3_**^−^**), potassium (K), and Ca in the petiole sap of plants subjected to treatment with both MDX-covered and non MDX-covered nTiO_2_ all exhibited increases (Table 4). However, only the concentration of NO_3_**^−^** demonstrated a further increase when nTiO_2_ was coated with MDX (Table 4). The observed increases in NO_3_^−^ and K exhibited a correlation with various plant responses, including an increase in fruit yield that corresponded to higher concentrations of both ions (Figure 5).

### 2.6. Effect of nTiO_2_ on Macronutrient Status in Fruits

The fruits obtained from plants that were treated with MDX-coated nTiO_2_ had the largest concentration of P. Based on the results of orthogonal contrasts, it was seen that the application of nTiO_2_ led to a further rise in P in the fruits, particularly when the NPs were surface-coated with MDX administered by drenching (Table 5). The application of MDX-covered nTiO_2_ resulted in an increase in fruit K; however, this rise was only observed when the treatment was applied by foliar spray (Table 5).

In comparison to the findings reported in the petiole sap, the application of non-MDX covered nTiO_2_ by foliar spray resulted in a minor but statistically significant drop in Ca levels in fruits, while magnesium (Mg) and N were unaffected (Table 5). The transformation of nutrient status data from concentration units to total nutrient content units revealed that the application of nTiO_2_ through foliar sprays led to an increase in N and Mg (Figure 6). Additionally, both foliar spraying and drench treatment resulted in higher P levels (Figure 6).

### 2.7. Effect of nTiO_2_ on Micronutrient and Titanium Status in Fruits

The use of MDX to cover nTiO_2_ resulted in a decrease in Fe levels in fruits, as demonstrated by orthogonal contrasts, and this drop was more pronounced when the NPs were applied by drenching (Table 6). Zinc demonstrated comparable patterns to Fe, and the decline was more pronounced in the plants exposed to non-MDX-covered nTiO_2_ (Table 6). Boron (B) exhibited no significant changes based on orthogonal comparisons, while B in fruits from plants treated with MDX-covered nTiO_2_ was observed to decrease (Table 6). Compared to the control plants, copper (Cu) and manganese (Mn) did not exhibit any significant changes in response to the application of nTiO_2_ regardless of the method of application (Table 6).

The impact of foliar nTiO_2_ on total nutrient content was calculated by adjusting the nutrient concentration according to the fruit yield on a dry basis. The results indicated that there was no significant influence on the levels of Fe and Zn, while there was a tendency for B and Cu to rise when plants were treated with foliar nTiO_2_ (Figure 7). The analysis of Ti concentration (Table 6) and content (Figure 7) indicates that treatment with nTiO_2_ did not have significant effects.

## 3. Discussion

### 3.1. Effect of nTiO_2_ on Fruit Production and Quality

Although there was no increase in the accumulation of Ti, the application of nano-sized TiO_2_ had a favorable impact on the production, quality, and size of the fruits. The observed responses could possibly be attributed to the favorable characteristics of Ti when utilized at a nanoscale level, mostly due to its gradual release mechanism, which ensures sustained availability over an extended period of time. The observed increase in crop yield resulting from the application of nTiO_2_ surface covering can be attributed to the effects of MDX. These effects include improved dispersibility of the nanoparticles due to the stabilizing and slow-release properties of MDX [25,26]. Additionally, MDX facilitates the formation of a solid network of carbohydrates linked by hydrogen bonds, which provides protection to nTiO_2_ against chemical degradation and alteration caused by other metal ions [25,27].

The findings of our study are consistent with previous research conducted on wheat (*Triticum aestivum* L.) [28] and coriander (*Coriander sativum* L.) [29]. In these studies, it was shown that application of nTiO_2_ at concentrations of 100 mg L^−1^ and 6 mg L^−1^ resulted in yield increases of 39% and 41%, respectively. Additional research has corroborated the positive impact of nTiO_2_ on crop yields. For instance, Zheng et al. [12] conducted a study which revealed that application of nTiO_2_ resulted in a significant increase in spinach yield ranging from 63% to 76% compared to control plants, whereas Mattiello and Marchiol [14] and Ahmad et al. [16] respectively observed notable increases in barley and mint yields when nTiO_2_ was applied at a concentration of 500 mg L^−1^ in the former case and 50 mg L^−1^ in the latter.

The observed enhancement in fruit firmness observed in the current investigation could potentially be linked to the scavenging function of TiO_2_ on ethylene [30,31], the hormone responsible for regulating the ripening process in fruits. This is supported by previous research demonstrating that films incorporating chitosan–TiO_2_ nanocomposites can effectively delay the decline in firmness in cherry tomatoes [30]. The improvement in fruit flavor, as demonstrated by our findings on the increase in total soluble solids, is supported by previous research indicating that the application of foliar sprays of elemental Ti during the yellow fruit stage of raspberries (*Rubus idaeus* L.) led to an average increase of 4.9% in this parameter [32]; nonetheless, there are other reports indicating that there is no effect of Ti in cherry tomatoes [33].

### 3.2. Effect of nTiO_2_ on Plant Growth

Previous studies have shown that nTiO_2_ can positively impact the growth and development of plants. However, there exists a degree of controversy associated with its effects due to the lack of investigation into certain factors, such as particle size, concentration, surface properties, form of application, and surface-covering with polysaccharides, which may have the potential to influence the resulting effects [11,34]. The results of our study coincide with research conducted by Rafique et al. [35], in which they observed a rise in dry biomass in wheat when subjected to a soil concentration of 60 mg kg^−1^ nTiO_2_. Even higher concentrations (80 and 100 mg kg^−1^) affected the total biomass in addition to the length of roots and shoots.

### 3.3. Effect of nTiO_2_ on SPAD and Gas Exchange

Our findings indicate that the application of nTiO_2_ to plants, independent of the method of application and whether the surface of the NPs was covered or not, led to an increase in the SPAD index. This shows that the presence of Ti contributed to an increase in chlorophyll content in the leaves, as reported by Yamamoto et al. [36] and Hu [37]. These results of our study coincide with previous research. For instance, in the case of barley, a 14% increase in the SPAD index was observed with foliar applications of nTiO_2_ at a concentration of 2000 ppm [38]. Similarly, in *Paeonia suffruticosa* Andrews, an application of 500 ppm resulted in a substantial increase of 84.7% [39], while Singh and Kumar [40] documented similar trends in spinach. The results of our study indicate that the exposure of plants to nTiO_2_ results in a positive impact on the photosynthetic apparatus. This is evidenced by the enhancement of chlorophyll content, as indicated by the higher SPAD index and the observed correlation between the SPAD index and photosynthesis, further supporting the beneficial effects of nTiO_2_ exposure on plant physiology. According to a study conducted by Morteza et al. [18], the utilization of nTiO_2_ enhances the synthesis of pigments and facilitates the conversion of light energy into active electrons and chemical reactions, leading to augmentation of photosynthetic efficiency.

The observed augmentation in the rate of photosynthesis may have led to elevated synthesis of carbohydrates, providing a plausible explanation for the observed rise in yield and buildup of biomass. The observed results provide support for the assertion made by previous researchers [41,42,43] that nTiO_2_ has a positive impact on the photosynthetic process. Mingyu et al. [42] have shown that the impact of anatase nTiO_2_ on photosynthesis can be attributed to two main factors, namely, enhanced absorption of light within the chloroplasts and the facilitation of excitation energy transfer to photosystem II. Other studies have shown that nTiO_2_ enhances the photosynthetic rate, concentration of photosynthetic pigments, ribulose bisphosphate carboxylase/oxygenase activity, electron transfer, and absorption and transport of light [12,15,16,18,44]. The observed augmentation in photosynthetic activity in plants treated with nTiO_2_ has been shown to be correlated with a reduction in internal CO_2_ content, likely attributable to an elevated rate of CO_2_ fixation during photosynthesis. The findings of this study align with those presented by Qi et al. [15], wherein the use of nTiO_2_ at concentrations ranging from 0.05 g L^−1^ to 0.2 g L^−1^ resulted in a reduction of internal CO_2_ concentration compared to control plants that received just pure water. The application of nTiO_2_ resulted in an increase in stomatal conductance in tomato plants, as observed in previous studies conducted by Ahmad et al. [16] and Tighe-Neira et al. [17]; specifically, the stomatal conductance increased by 8.5% in mint and radish plants when exposed to nTiO_2_. Similar trends were reported by Gao et al. [45] in seedlings of elm (*Ulmus elongate* LK Fu & CS Ding).

### 3.4. Effect of nTiO_2_ on Macronutrient Status

Regarding the impact of nTiO_2_ on mineral content, Alharby et al. [46] observed comparable findings to ours in wheat. They found that the application of nTiO_2_ led to an increase in P and K in both the shoots and roots, specifically when nTiO_2_ was sprayed at a concentration of up to 400 mg kg^−1^ of soil. The findings of Servin et al. [47] showed similar results, with the levels of P and K increasing in cucumber fruits (*Cucumis sativus* L.) from plants subjected to a soil treatment of 500 mg kg^−1^ with nTiO_2_. The researchers suggested that this phenomenon may be attributed to the influence of nTiO_2_ on plant hormones such as cytokinins and gibberellins. The application of nTiO_2_ has been shown to result in a drop in Ca and Mg in the leaves of fenugreek (*Trigonella foenum-graecum* L.), which is linked to the reduced absorption and distribution of essential minerals [48]. In the present study, we propose that the observed results, namely, the absence of a significant impact of nTiO_2_ on N and Mg concentration in the fruits as well as the reduction in Ca, might be attributed to a dilution effect. This dilution effect may arise from the necessity of distributing the nutrients absorbed throughout the greater biomass of the fruits resulting from the enhanced yield. The conversion of nutrient status data from concentration units to total nutrient content units revealed an increase in N and Mg concentrations in tomato treated with nTiO_2_ through foliar spraying. Additionally, both the foliar spraying and drenching applications resulted in higher P levels.

### 3.5. Effect of nTiO_2_ on Micronutrient Status

The results relative to the concentration of micronutrients in our study differ from those reported in prior research conducted on oats (*Oryza sativa* L.) [49] and wheat [46]. An increase in Fe content was seen after the application of nTiO_2_ in both oats and wheat. In contrast to our findings, Mattiello and Marchiol [14] observed an elevation in the concentrations of Fe, Zn, and Mn in barley grains. Similarly, Alharby et al. [46] reported an increase in Fe and Mn in the shoots and roots as well as in the grain of wheat plants when exposed to nTiO_2_ at a soil concentration of up to 400 mg kg^−1^. Similarly, the use of nTiO_2_ particles with a size of 83 nm on fenugreek leaves resulted in elevated levels of Zn and Mn in both the leaves and stems; however, Fe and Cu decreased in comparison to the utilization of NPs with a size of 23 nm [48]. Our results indicate that there was a reduction in the concentrations of Fe, Zn, and B in the fruits of plants subjected to treatment with nTiO_2_. As observed for the macronutrients, this drop might be attributed to a dilution effect caused by the increased fruit production observed in plants that received such treatments. This observation arises when the nutrient concentration is adjusted in relation to the fruit yield, resulting in there being no significant impact on the content of Fe and Zn, while the content of B and Cu exhibited a tendency to increase when plants were subjected to foliar applications of nTiO_2_.

### 3.6. Effect of nTiO_2_ on Fruit Titanium

The findings of the present study indicate that the administration of Ti by leaf spraying or drenching treatments may not result in the translocation of Ti to the fruits. In accordance with our findings, Pošćić et al. [50] observed that the presence of nTiO_2_ at concentrations of 500 and 1000 mg kg^−1^ in soil did not provide a substantial increase in Ti content in barley kernels. However, it is worth noting that Servin et al. [47] previously demonstrated the existence of nano-sized nTiO_2_ in cucumber fruits while Kelemen et al. [51] have reposted that the application of Ti to the leaf surface results in its uniform translocation to both the leaves and roots. Although there is evidence that Ti absorbed through both leaves and roots can be translocated to different parts of the plant [11], the lack of observed impact on Ti concentration in fruits when nTiO_2_ is applied via drenching may be attributed to its preferential accumulation in the roots. Kelemen et al. [51] have reported that only a minimal quantity of Ti is transported to the shoots.

Several studies have suggested that the crystal structure of nTiO_2_ can impact the movement of Ti within different plant parts. For instance, Cai et al. [52] found that Ti in anatase form did not translocate from the roots to other parts of rice plants due to the presence of the Casparian strip, which acts as a barrier. On the other hand, Servin et al. [47,53] demonstrated that in cucumber the rutile form of nTiO_2_ was primarily detected in the aboveground parts, while the anatase form remained in the roots. One additional factor that could have influenced the absence of an effect on Ti concentration in the fruits is the particle size. Previous studies have shown that the use of larger-sized nanoscale TiO_2_ particles may lead to a reduction in Ti content in leaves, stems, and roots. This reduction occurs because the root cells are unable to absorb large-diameter NPs due to their inability to fit through the pore diameter of the cell wall. This phenomenon was demonstrated by Missaoui et al. [48] in their research on fenugreek.

## 4. Materials and Methods

### 4.1. Surface Covering of nTiO_2_

Nanoparticles of TiO_2_ were obtained from commercial sources of guaranteed grade (Sigma-Aldrich, Reference 799289, St. Louis, CA, USA, EE.UU.). The surface modification of nTiO_2_ was performed by the precipitation method [54] using MDX (Manuchar Inc., Monterrey, México). The modification procedure was as follows: in a volumetric flask (PYREX, Charleroi, PA, USA), 1.5 g nTiO_2_ and 1.5 g MDX were dissolved in 50 mL of ethanol (99.5%) with agitation. In a stirring plate (Thermo Scientific, Model SP194715, Waltham, MA, USA), a heating mantle (Glas-Col, Model 100AO399, Terre Haute, IN, USA) was set at 65 °C; the flask was inserted and a condenser coil (Synthware Glass Inc., model C321410, Beijing, China) was connected to cool the reaction vapors through a 4 °C refrigerated circulating bath (PolyScience, Model 9102, Niles, IL, USA) for 6 h starting at the boiling point. After 6 h the heating mantle was switched off, leaving the flask shaking in the refrigerant bath for 3 h. Subsequently, the solution from the flask was recovered in an 85 mL polypropylene centrifuge tube (Thermo Scientific, Model Nalgene 3118-0085, Waltham, MA, USA) and centrifuged for 10 min at 15,000 rpm (Fisher Scientific, model AccuSpin 3, Waltham, MA, USA), the supernatant was decanted, and the sediment washed with 50 mL of ethanol (99.5%) twice to remove residues. Finally, the nTiO_2_ sediment was placed in an oven (Thermo Scientific model OMH 180-S, Waltham, MA, USA) at 80 °C for 24 h and the NPs were recovered and stored in a glass vial (Thermo Scientific, Model Sterilin, Waltham, MA, USA).

### 4.2. Characterization of nTiO_2_

The crystalline structure and chemical stability of the NPs were determined by X-ray Diffraction (XRD) using a diffractometer in the range of 10 to 80° in 2 Ɵ (CuKα, 25 mA, 35 kV) (Siemens model D-500, Berlin, Germany). First, 1 g of nTiO_2_ (dry powder) was placed in the sampling port and evenly distributed; the sampling port was then attached inside the diffractometer. The diameter, morphology, and size distribution of the nTiO_2_ were determined via high-resolution transmission electron microscopy (HRTEM FEI, model TITAN 80-300, Hillsboro, OR, USA). A drop of the resulting mixture was placed in a grid for transmission electron microscopy analysis and allowed to dry before proceeding to observation under the microscope. Detection of the functional groups of the nTiO_2_ was performed with the infrared spectroscopy technique (FT-IR) using a spectrometer (Thermo Scientific Model Nicolet iS50, Waltham, MA, USA). To complete the analysis, the 1 g sample of nTiO_2_ was placed in the sampling port and the diamond tip was placed over the samples to proceed with the readings.

### 4.3. Study Site and Growing Conditions

The experiment was carried out in a greenhouse at Universidad Autónoma Agraria Antonio Narro (25°23′42″ N Lat., 100°59′57″ W Long., 1743 m above sea level). Environmental data were measured with a data logger (Watch Dog 1000 Series, Spectrum Technologies, Inc. Aurora, IL, USA). The temperature during the study was 23.2 °C (mean maximum), 12.4 °C (mean minimum), and 18.1 °C (seasonal average), while the relative humidity was 95% (mean maximum), 47% (mean minimum), and 72% (season average). The mean seasonal photosynthetically active radiation was 370 µmol m^–2^ s^–1^.

### 4.4. Plant Material and Experimental Conduction

Tomato cv. Climstar seeds were sown on 13 February 2018 in 1 L containers. Four weeks later, the plants (20 cm length and three fully developed leaves) were transplanted into 12 L containers with a mixture of sphagnum peat (PREMIER, Premier Tech, Toronto, ON, Canada) (70% *v/v*) and horticultural grade perlite (Hortiperl, Monterrey, México) (30% *v/v*). Initial medium pH and electrical conductivity (EC) were 5.7 (adjusted with sodium bicarbonate (1.0 g L^−1^) and 0.8 dS m^−1^, respectively.

Plants were irrigated with a complete nutrient solution [55] containing (meq L^−1^): 12 NO_3_^−^, 1 H_2_PO_4_^−^, 7 K, 9 Ca, 4 Mg, and 7 SO_4_^2−^. Micronutrients were provided at (mg∙L^−1^): 5.3 Fe-EDTA, 0.4 Zn-EDTA, 2.6 Mn-EDTA, 0.5 Cu-EDTA, 0.2 B (Na_2_[B_4_O_5_ (OH)_4_] ·8H_2_O), and 0.2 Mo (Na_2_MoO_4_). Nutrient solution pH (Hanna Combo HI 98129, Hanna Instruments, Inc., Smithfield, RI, USA) was adjusted to 6.0 ± 0.1 (using 4.5 meq L^−1^ HNO_3_ and 1.0 meq L^−1^ H_3_PO_4_ in the preparation) and EC (Hanna Combo HI 98129, Hanna Instruments, Inc., USA) at 2.3 dS m^−1^ and was applied though an automated fertigation system consisting of two drip irrigation emitters (Netafim Irrigation Inc., Fresco, CA, USA) per container dispensing 1 L·h^–1^ each. During the vegetative phase, three irrigations of 2 h each were applied per week for a total of 4 L per plant every other day, while in the reproductive phase ~3.5 h irrigations were applied on a daily basis for a total of 7 L every day. The leaching fraction was determined by incremental additions of nutrient solution until leachate was observed on the bottom of the container, collected, and measured to determine the solution retained by the substrate, and was maintained at ~35% throughout the experiment.

Plants were separated 45 cm between containers from center to center and 120 cm between rows of containers, for a total density of 3 plants m^−2^. The plants were trellised to one stem by eliminating all the lateral shoots when they were 5 cm in length. The leaves were pruned periodically throughout the study period by eliminating the older leaves at the bottom part of the plants in order to maintain 11 to 13 mature leaves; trusses were pruned to maintain five flowers per truss, and ten trusses were allowed to develop during the study.

### 4.5. Foliar and Drench Applications of nTiO_2_

Two foliar (at 40 and 100 days after sowing) or drench applications (at 1 and 100 days after sowing) were manually conducted with 100 mL of nTiO_2_ solution at 0 or 1500 mg L^−1^. Foliar sprays were carefully applied, covering the plants with a polyethylene sheet to avoid the spray reaching other plants with no treatment. The nTiO_2_ solutions, previously dispersed with a sonicator (Sonics model VC505, Newtown, CT, USA) during 15 min at 38% absorbance, were prepared with either MDX-surface-covered or no MDX-surface covered NPs. In addition to these four treatments (foliar with MDX, foliar with no MDX, drenching with MDX, and drenching with no MDX), there was a control treatment in which a control solution with no nTiO_2_ was used. All the solutions had a final pH = 5.6–5.8 at application time.

### 4.6. Fruit Yield and Quality

The total yield of tomato fruits was calculated by adding the weight of the fruits from ten trusses harvested during 77 days using a digital scale (Rhino, model BAPRE-3, CHN). In the five fruits from the tenth truss, firmness was measured as the resistance to puncture with a penetrometer (AKSO model FT327, 8 mm tip) in four areas around the equatorial part of the fruit and averaged. The juice from tomato fruits was extracted by squeezing the fruit to obtain a few drops; the drops were placed onto the refractometer prism plate and the total soluble solids measured with a refractometer (ATAGO model ATC-1E).

### 4.7. Plant Growth, Dry Weight and SPAD

Plant height and basal stem diameter were measured at the end of the experiment with a tape measure and a caliper. After the final fruit harvest, the plants were collected, bagged, and placed in an oven (Novatech, model HS45-AIA, Murrieta, CA, USA) at 70 °C for 72 h. The total dry weight of the plant was recorded, including pruned leaves, leaves, stems, and roots. The SPAD index (SPAD-502 meter, Minolta, Japan) was recorded at intervals of 15 days throughout the duration of the study. Two healthy recently mature leaves were selected per experimental unit, and on each leaf three randomly chosen spots were sampled to obtain the SPAD value.

### 4.8. Gas Exchange Parameters

The photosynthetic rate, stomatal conductance, intracellular CO_2_, and transpiration rate were measured with a portable photosynthesis system (LI-COR 6400XT, Biosciences, Lincoln, NE, USA) in leaves from the middle part of the plant. Measurements were conducted three times every 28 days between 11 am and 1 pm and the data were averaged. During the measurements, environmental conditions in the cuvette were held at 375.8 µmol mol^−1^ CO_2_ concentration, air temperature was held at 33.2 °C, and the relative humidity was held at 54.9%. Measurements were conducted four times on each sampled leaf.

### 4.9. Fruit Nutrient Status

Fruits from the tenth truss were dried in an oven at 70 ° C for 72 h (Novatech, model HS45-AIA, Murrieta, CA, USA) and then ground to pass a 40-mesh screen (Tekmar, model A-10, Cole-Parmer Instrument, Chicago, IL, USA) for the determination of nutrient concentration. Nitrogen was determined in 0.25 g ground samples following Kjeldahl’s procedure [56], while P, K, Ca, Mg, Fe, Zn, Cu, Mn, B, and Ti were determined in fruit tissues digested in a 2:1 mixture of H_2_SO_4_:HClO_4_ and 2 mL of 30% H_2_O_2_ using an inductively coupled plasma atomic emission spectrometer (ICP-AES) (model Liberty, VARIAN, Santa Clara, CA, USA) [57]. Nutrient content in the fruits was calculated considering the fruit nutrient concentration adjusted using the dry weight of all the fruits harvested throughout the experiment.

### 4.10. Petiole Sap Nutrient Status

Nitrate, K, and Ca concentrations were measured with portable ionmeters (LAQUAtwin, Horiba Scientific, Kyoto, Japan) in the sap extracted from petioles of fully mature leaves collected 90 days after transplant. The sap was extracted with a mortar when the leaves reached 24–26 °C and then stored at 10 °C until the readings were taken.

### 4.11. Experimental Design and Statistical Analysis

The five treatments selected for the study were set in a randomized block design; each treatment consisted of five one-plant replicates. The statistical analysis was conducted with ANOVA, and when significance was detected a multiple mean comparison test was performed with Duncan`s procedure (*p* = 0.05) using SAS 9.0. Orthogonal contrasts were conducted to determine the significance of the effect between nTiO_2_ or MDX-nTiO_2_ compared to the control plants and between MDX-treated and non MDX-treated nTiO_2_.

## 5. Conclusions

The utilization of nano-sized TiO_2_ on tomato plants led to an increase in the SPAD index, suggesting a rise in the concentration of chlorophyll. The observed augmentation in chlorophyll concentration demonstrated a positive correlation with the rate of photosynthesis, resulting in improved plant growth and higher yields in tomato plants. In addition, the application of nano-sized TiO_2_ resulted in an enhancement of the nutritional composition of petiole sap along with elevated levels of P and K in fruits. However, the utilization of nano TiO_2_ led to a reduction in the amounts of Fe, Zn, and B. In this study, we observed a dilution effect in connection to the nutrient composition of the fruits, particularly in relation to the rising levels N, P, Mg, and Cu. No significant changes in the Ti content of the fruits were noticed. The use of MDX as a surface coating demonstrated enhanced response of the plants towards nTiO_2_, leading to increased fruit yield and higher quality in comparison to plants exposed to uncoated nTiO_2_. In many cases, the application of drench treatments using MDX-coated nTiO_2_ resulted in a more advantageous plant response.

## Figures and Tables

**Figure 1 plants-12-03099-f001:**
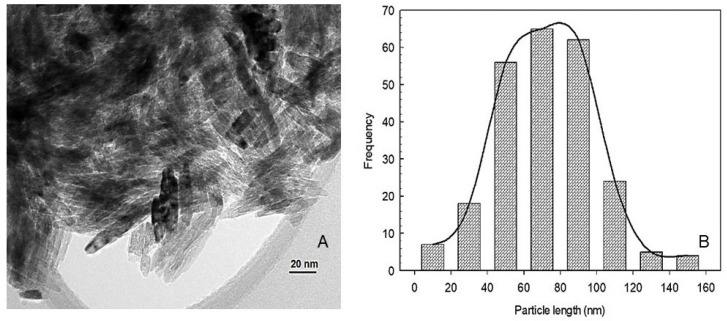
Micrograph of nanoparticles of TiO_2_ observed through electron transmission microscope (**A**) and nanoparticle size distribution (**B**).

**Figure 2 plants-12-03099-f002:**
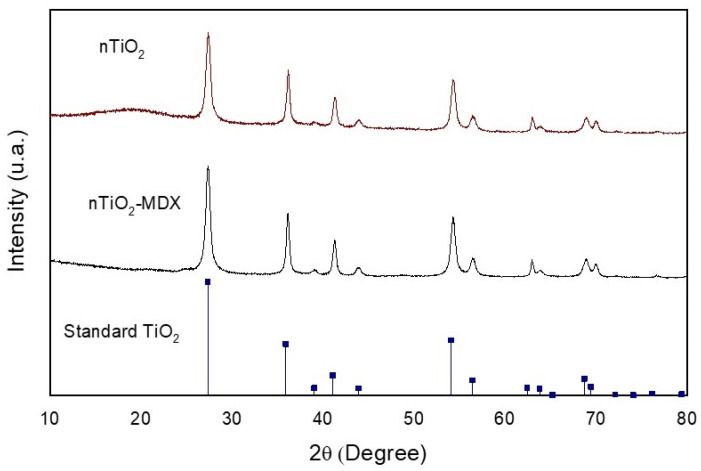
X-ray diffractogram of nTiO_2_ with no surface covering, maltodextrin (MDX) surface-covered nTiO_2_, and rutile TiO_2_ standard (JCPDS 01-077-0443).

**Figure 3 plants-12-03099-f003:**
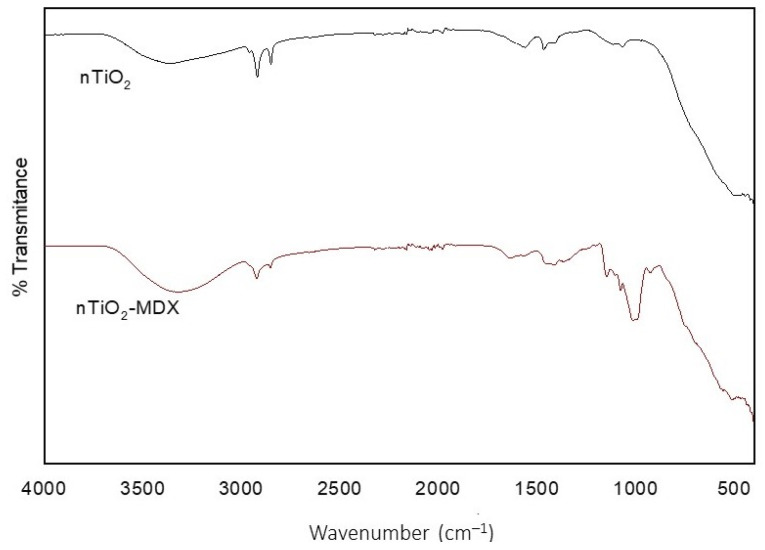
Fourier-transformed infrared spectra of nTiO_2_ with no surface covering and with maltodextrin (MDX) surface covering_._

**Figure 4 plants-12-03099-f004:**
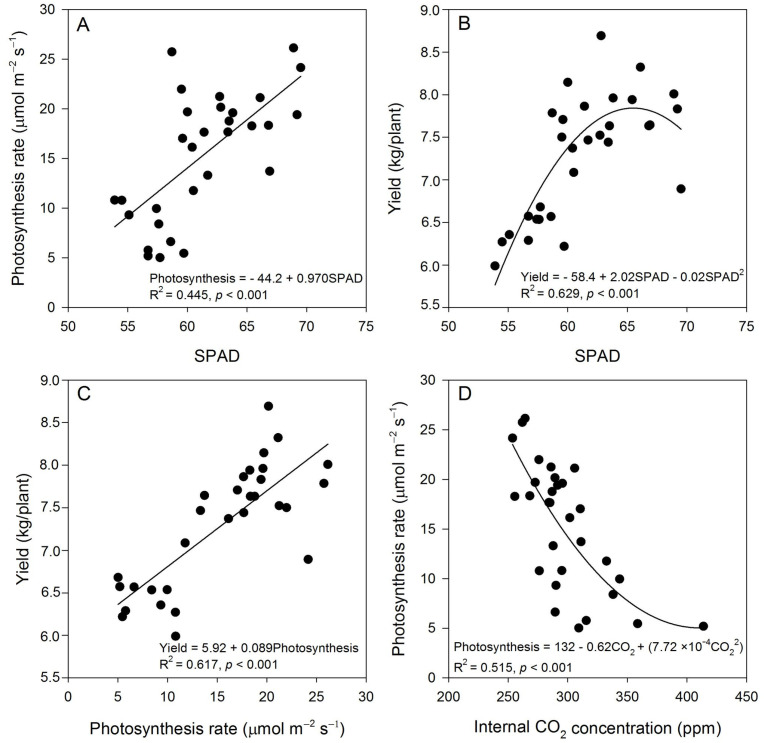
Relationship among gas exchange parameters and SPAD index with fruit yield of tomato plants treated with covered or non-maltodextrin-covered nTiO_2_. Photosynthesis rate (**A**) and fruit yield (**B**) as affected by the SPAD index, and the relationship between photosynthesis and yield (**C**) and between the internal CO_2_ concentration and photosynthesis (**D**).

**Figure 5 plants-12-03099-f005:**
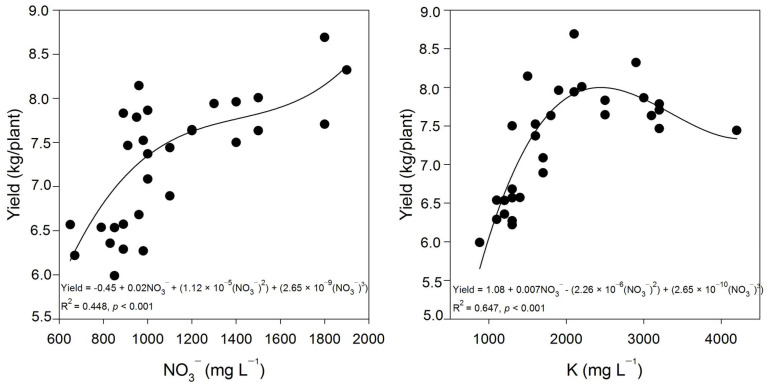
Relationships of NO_3_**^−^** and K concentration within the petiole sap and fruit yield of tomato plants treated with covered or non-maltodextrin-covered nTiO_2_.

**Figure 6 plants-12-03099-f006:**
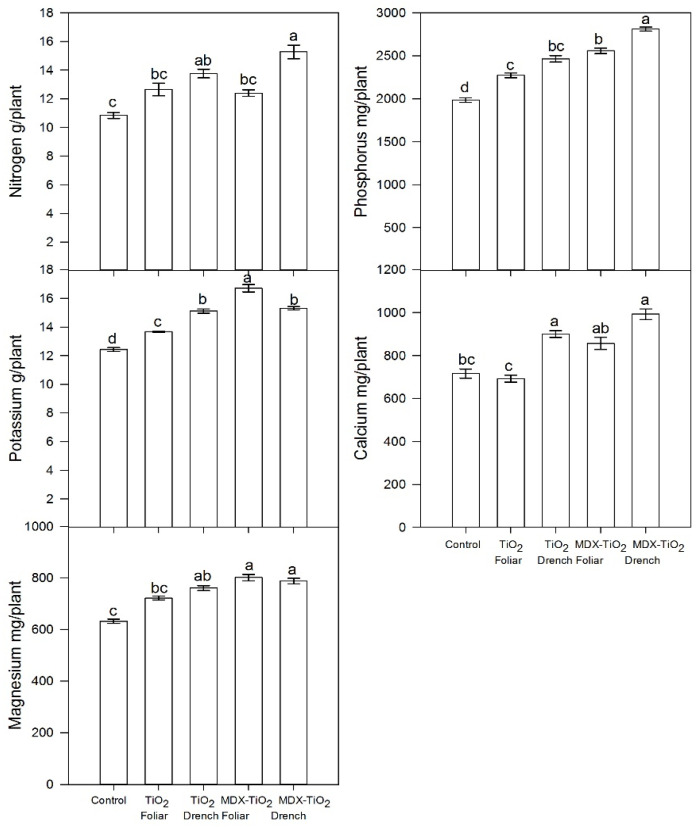
Effects of foliar and drench applications of nTiO_2_ at 1500 ppm on fruit macronutrient content adjusted by the dilution effect. nTiO_2_ was treated with or without maltodextrin (MDX) surface covering. Columns with different letters indicate significant differences according to Duncan’s test at *p* = 0.05. Bars represent the standard error of the mean.

**Figure 7 plants-12-03099-f007:**
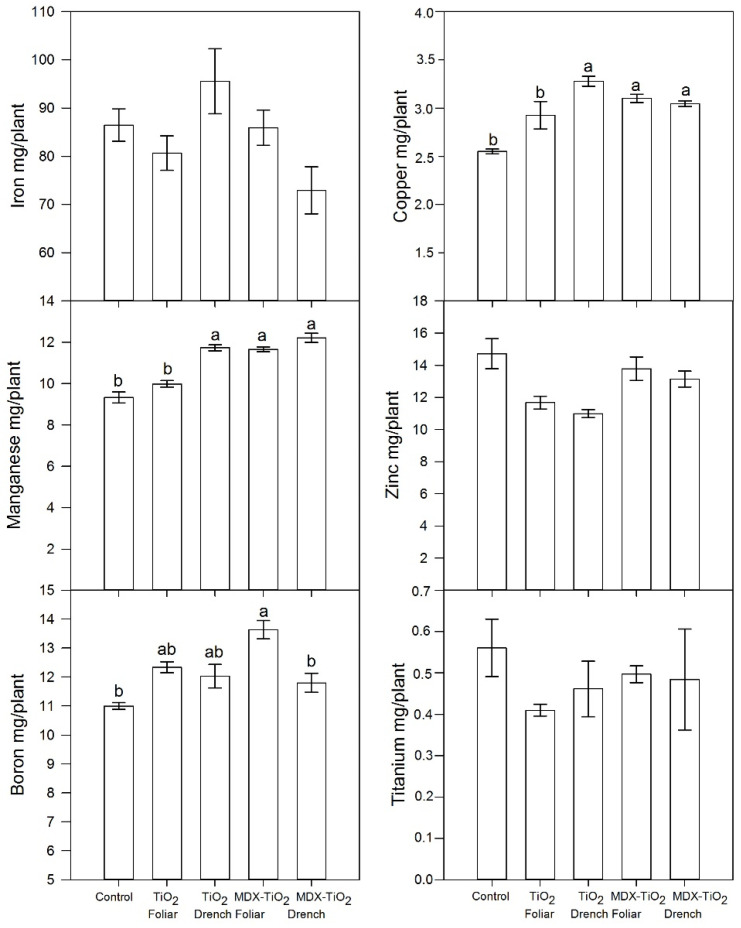
Effects of foliar and drenching applications of nTiO_2_ at 1500 ppm on fruit micronutrient content adjusted by the dilution effect. nTiO_2_ was treated with or without maltodextrin (MDX) surface covering. Columns with different letters indicate significant differences according to Duncan’s test at *p* = 0.05. Bars represent the standard error of the mean.

**Table 1 plants-12-03099-t001:** Effects of foliar and drench applications of nTiO_2_ at 1500 ppm on the attributes of tomato fruits. nTiO_2_ was treated with or without maltodextrin (MDX) surface covering.

Treatment	Application Method	Yield kg/Plant	Fruit Weight g	PD mm	ED mm	Firmness kg m^−2^	TSS °Brix
Control		6.40 c	128.0 c	52.6 c	63.8 c	2.96 b	5.09 b
nTiO_2_	Foliar	7.57 b	151.4 b	55.0 ab	66.6 ab	3.25 b	5.40 b
	Drench	7.41 b	148.2 b	54.1 b	66.1 b	3.28 b	6.66 a
nTiO_2_ + MDX	Foliar	7.77 b	155.5 b	54.9 ab	67.9 a	3.94 a	6.70 a
	Drench	8.14 a	162.8 a	55.6 a	67.6 a	3.21 b	6.94 a
	ANOVA	*p* = 0.001	*p* = 0.001	*p* = 0.001	*p* = 0.001	*p* = 0.001	*p* = 0.001
Orthogonal contrasts							
Control vs. nTiO_2_		*p* = 0.001	*p* = 0.001	*p* = 0.001	*p* = 0.001	p = 0.025	*p* = 0.001
Control vs. nTiO_2_-MDX		*p* = 0.001	*p* = 0.001	*p* = 0.001	*p* = 0.001	*p* = 0.001	*p* = 0.001
nTiO_2_ vs. nTiO_2_-MDX		*p* = 0.001	*p* = 0.002	ns	*p* = 0.003	*p* = 0.025	*p* = 0.001

PD = Polar Diameter, ED = Equatorial Diameter, TSS = Total Soluble Solids. Means followed by different letters within the same column indicate significant differences according to Duncan’s comparison test (*p* = 0.05). ns = non-significant.

**Table 2 plants-12-03099-t002:** Effects of foliar or drench applications of nTiO_2_ at 1500 ppm on the growth, dry weight (DW), and SPAD index of vegetative plant parts. nTiO_2_ was treated with or without maltodextrin (MDX) surface covering.

Treatment	Application Method	Plant Height m	Stem Diameter Mm	Leaf DW g	Root DW g	Stem DW g	Total DW g	SPAD
Control		2.10 b	20.3 c	162 b	22.3 b	89.7 c	274 c	56.8 b
nTiO_2_	Foliar	2.25 ab	21.8 b	193 b	26.0 b	109.2 ab	328 b	63.6 a
	Drench	2.25 ab	24.5 a	178 b	25.6 b	96.6 ab	300 bc	62.6 a
nTiO_2_- + MDX	Foliar	2.34 a	22.8 b	165 b	21.8 b	95.4 bc	282 c	63.7 a
	Drench	2.25 ab	25.4 a	226 a	36.4 a	111.8 a	374 a	64.2 a
	ANOVA	*p* = 0.031	*p* = 0.001	*p* = 0.001	*p* = 0.001	*p* = 0.009	*p* = 0.001	*p* = 0.003
Orthogonal contrasts								
Control vs. nTiO_2_		*p* = 0.017	*p* = 0.001	*p* = 0.047	*p* = 0.037	*p* = 0.017	*p* = 0.001	*p* = 0.001
Control vs. nTiO_2_-MDX		*p* = 0.003	*p* = 0.001	*p* = 0.006	*p* = 0.001	*p* = 0.013	*p* = 0.001	*p* = 0.001
nTiO_2_ vs. nTiO_2_-MDX		ns	*p* = 0.049	ns	*p* = 0.048	ns	ns	ns

Means followed by different letters within the same column indicate significant differences according to Duncan’s comparison test (*p* = 0.05). ns = non-significant.

**Table 3 plants-12-03099-t003:** Effects of foliar and drenching applications of nTiO_2_ at 1500 ppm on gas exchange parameters. nTiO_2_ was treated with or without maltodextrin (MDX) surface covering.

Treatment	Application Method	Photosynthesis Rate µmol CO_2_ m^−2^s^−1^	Stomatic Conductance mol H_2_O m^−2^s^−1^	Internal Concentration μmol CO_2_ mol air^−1^	Transpiration Rate mmol H_2_O m^−2^s^−1^
Control		7.74 b	0.390 b	323.0 a	7.59 b
nTiO_2_	Foliar	17.05 a	0.461 b	296.0 ab	8.21 b
	Drench	20.11 a	0.450 b	278.2 b	8.94 ab
nTiO_2_ + MDX	Foliar	18.40 a	0.439 b	276.7 b	9.43 ab
	Drench	20.81 a	0.600 a	293.1 ab	10.40 a
	ANOVA	*p* = 0.001	*p* = 0.003	*p* = 0.037	*p*=0.013
Orthogonal contrasts					
Control vs. nTiO_2_		*p* = 0.001	*p* = 0.045	*p* = 0.011	*p* = 0.050
Control vs. nTiO_2_-MDX		*p* = 0.001	*p* = 0.017	*p* = 0.008	*p* = 0.001
nTiO_2_ vs. nTiO_2_-MDX		ns	*p* = 0.049	ns	*p* = 0.015

Means followed by different letters within the same column indicate significant differences according to Duncan´s comparison test (*p* = 0.05). ns = non-significant.

**Table 4 plants-12-03099-t004:** Effects of foliar and drenching applications of nTiO_2_ at 1500 ppm on petiole sap nutrient concentration. nTiO_2_ was treated with or without MDX surface covering.

Treatment	Application Method	NO_3_^−^ mg L^−1^	K mg L^−1^	Ca mg L^−1^
Control		836 c	1208 b	581 c
nTiO_2_	Foliar	1098 b	2640 a	844 ab
	Drench	988 bc	2260 a	942 ab
nTiO_2_ + MDX	Foliar	1212 b	2100 a	744 bc
	Drench	1680 a	2460 a	1038 a
	ANOVA	*p* = 0.001	*p* = 0.004	*p* = 0.001
Orthogonal contrasts				
Control vs. nTiO_2_		*p* = 0.009	*p* = 0.003	*p* = 0.001
Control vs. nTiO_2_-MDX		*p* = 0.001	*p* = 0.001	*p* = 0.001
nTiO_2_ vs. nTiO_2_-MDX		*p* = 0.001	ns	ns

Means followed by different letters within the same column indicate significant differences according to Duncan´s comparison test (*p* = 0.05). ns = non-significant.

**Table 5 plants-12-03099-t005:** Effects of foliar and drench applications of nTiO_2_ at 1500 ppm on fruit macronutrient concentration. nTiO_2_ was treated with or without maltodextrin (MDX) surface covering.

Treatment	Application Method	N %	P mg kg^−1^	K mg kg^−1^	Ca mg kg^−1^	Mg mg kg^−1^
Control		2.12 a	3875.6 bc	24280.4 bc	1398.8 a	1233.8 a
nTiO_2_	Foliar	2.09 a	3752.9 c	22569.3 c	1144.0 b	1192.3 a
	Drench	2.32 a	4158.9 ab	25489.1 ab	1519.2 a	1283.1 a
nTiO_2_ + MDX	Foliar	1.99 a	4116.0 abc	26880.3 a	1378.4 a	1289.4 a
	Drench	2.35 a	4315.5 a	23516.7 c	1524.8 a	1209.8 a
	ANOVA	*p* = 0.200	*p* = 0.028	*p* = 0.003	*p* = 0.002	*p* = 0.325
Orthogonal contrasts						
Control vs. nTiO_2_		ns	*p* = 0.002	ns	ns	ns
Control vs. nTiO_2_-MDX		ns	*p* = 0.003	ns	ns	ns
nTiO_2_ vs. nTiO_2_-MDX		ns	*p* = 0.006	ns	ns	ns

Means followed by different letters within the same column indicate significant differences according to Duncan´s comparison test (*p* = 0.05). ns = non-significant.

**Table 6 plants-12-03099-t006:** Effects of foliar and drenching applications of nTiO_2_ at 1500 ppm on fruit micronutrient concentration. nTiO_2_ was treated with or without maltodextrin (MDX) surface covering.

Treatment	Application Method	Fe mg kg^−1^	Cu mg kg^−1^	Mn mg kg^−1^	Zn mg kg^−1^	B mg kg^−1^	Ti mg kg^−1^
Control		168.9 a	4.98 ab	18.2 ab	28.7 a	21.5 a	0.847 a
nTiO_2_	Foliar	133.2 ab	4.83 b	16.5 b	19.3 b	20.4 ab	0.677 a
	Drench	161.2 a	5.53 a	19.8 a	18.5 b	20.3 ab	0.778 a
nTiO_2_ + MDX	Foliar	138.2 ab	4.99 ab	18.8 ab	22.2 ab	21.9 a	0.799 a
	Drench	112.0 b	4.68 b	18.8 ab	20.2 b	18.1 b	0.743 a
	ANOVA	*p* = 0.028	*p* = 0.151	*p* = 0.257	*p* = 0.014	*p* = 0.005	*p* = 0.942
Orthogonal contrasts							
Control vs. nTiO_2_		ns	ns	*p* = 0.019	ns	ns	*p* = 0.042
Control vs. nTiO_2_-MDX		*p* = 0.002	ns	*p* = 0.039	*p* = 0.001	ns	*p* = 0.050
nTiO_2_ vs. nTiO_2_-MDX		*p* = 0.049	ns	ns	*p* = 0.025	ns	ns

Means followed by different letters within the same column indicate significant differences according to Duncan´s comparison test (*p* = 0.05). ns = non-significant.

## Data Availability

Data are contained within the article.

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
