# Peer review of "Covered Rutile-TiO2 Nanoparticles Enhance Tomato Yield and Growth by Modulating Gas Exchange and Nutrient Status"

_plants, 2023, doi:10.3390/plants12173099_

Round 1
Reviewer 1 Report
The manuscript entitled “Covered Nanoparticles of Rutile-TiO2 Promote Growth and Yield of Tomato by Affecting Gas Exchange and Nutrient Status” aimed to investigate the effects of titanium dioxide nanoparticles (nTiO2) covered with maltodextrin (MDX) on tomato growth and yield. The topic of this experimental paper is well chosen and within the scope of the journal. The conclusion is supported by the obtained data. However, it is subjected to improvement before it is acceptable for publication with due attention to the following weakness.
1. Line 28: Too many keywords. Please delete the unimportant keywords and add the keyword “titanium dioxide nanoparticles”.
2. Line 55: Need to change the expression of “3.13×”.
3. Line 177: The unit of P, K, Ca and Mg concentration in fruit should change to mg/kg.
4. Line 199: The unit in Table 6 should be mg/kg.
5. Line 359: The first appearance of “Amp” should use the full name.
Author Response
We express our gratitude for the valuable remarks and recommendations provided by Reviewer 1. Our responses to the comments stated by him/her is as follows:
1. Line 28: Too many keywords. Please delete the unimportant keywords and add the keyword “titanium dioxide nanoparticles”.
REPLY TO COMMENT 1: According to the Journal´s Instructions for authors, 3 to 10 keywords are considered valid. We included 7 keywords. However, accepting the suggestion made by the Reviewer we eliminated some unimportant keywords and included the one suggested
2. Line 55: Need to change the expression of “3.13×”.
REPLY TO COMMENT 2. The sentence was rephrased to: “…while the photosynthetic rate exhibited a 3.13-fold increase [12].”
3. Line 177: The unit of P, K, Ca and Mg concentration in fruit should change to mg/kg.
REPLY TO COMMENT 3. The units on Table 4 and 5 were changed to “mg L‒1” or “mg kg‒1”
4. Line 199: The unit in Table 6 should be mg/kg.
REPLY TO COMMENT 4. The units on Table 6 was changed to “mg kg‒1”
5. Line 359: The first appearance of “Amp” should use the full name.
REPLY TO COMMENT 5. "Amp" was substitued by "Amplitude"

Reviewer 2 Report
comments

Author Response
We express our gratitude for the valuable remarks and recommendations provided by Reviewer 2. Our responses to the comments stated by him/her is as follows:
1.- Why you didn’t Consider Environmental Factors in your experiment? It was shall to Take note of environmental factors that could influence your results, such as weather conditions, soil properties, and other agronomic practices. These factors can impact the effectiveness of foliar and drench applications.
REPLY TO COMMENT 1: In the materials and methods section, there is a sub-section, "4.3 study site and growing conditions" on which we describe the environmental conditions. In Lines 352–356, we wrote, "…Temperature during the study was 23.2 °C (mean maximum), 12.4 °C (mean minimum), and 18.1 °C (seasonal average), while relative humidity was 95% (mean maximum), 47% (mean minimum), and 72% (season average). The mean seasonal photosynthetically active radiation was 370 µmol m–2 s–1." While in Lines 394-396 of the subsection "4.4 Plant material and experimental conduction" we described the substrate chemical properties (we did not use soil) as we wrote "…with a mixture of sphagnum peat (60% v/v) and perlite (40% v/v). Initial medium pH and electrical conductivity (EC) were 5.7 and 0.8 dS m‒1, respectively."
2.- It is not clear how to irrigate, the number of irrigations, and the duration between irrigations.
REPLY TO COMMENT 2: In Section 4.4, we detailed the utilization of two emitters that dispensed 1 liter of nutrient solution per hour for the purpose of irrigation. It was also noted that irrigations were administered three times each week, each lasting for a length of two hours, during the vegetative phase. Consequently, each irrigation amounted to four liters per plant. During the period of fruit production, daily irrigations were administered, with each irrigation lasting for a length of 3.5 hours. Consequently, each irrigation amounted to a volume of 7 liters per plant. In order to minimize any potential misunderstandings, the section was restated in the following manner: “During the vegetative phase, three 2–h irrigations were applied per week (a total of 4 L per plant every other day), while in the reproductive phase ~3.5 h irrigations were applied on a daily basis (a total of 7 L every day).”
3.- In Measuring Yield and Quality: If one of your objectives is to improve yield or crop quality, in adding measuring these parameters like Collect data on yield per unit area and assess the quality of harvested produce, such as size, color also you have to measure flavor, and shelf life.
REPLY TO COMMENT 3: We concur with the reviewer's assertion that the size, color, taste, and shelf life of tomato fruits are crucial quality characteristics. In the present investigation, we present results in Table 1, which illustrate the impact of the treatments on the weight, polar diameter, and equatorial diameter; we consider that these parameters serve as indicators of fruit size. Regarding the aspect of taste, we incorporated an fruit parameter known as "total soluble solids". This parameter serves as an indicator of the sugar concentration in fruits and is widely recognized as one of the key factors influencing fruit flavor. Apologies for the omission of criteria pertaining to the assessment of color and shelf life in our study. Nevertheless, we did incorporate the measurement of fruit firmness, which serves as an additional sign for determining the post-harvest storage duration of fruits. The suggestion made by the reviewer to include "yield per area" has not been incorporated into our work. This decision is based on the assumption that readers who are interested in this particular metric may readily compute it themselves. In subsection 4.4, it was specified that a plant density of 3.0 plants per square meter was employed. Consequently, to ascertain the yield per square meter, one just has to multiply the average yield per plant by 3.0, and this calculation may be extended to encompass greater areas.
4.- Also its very important to focus, regularly monitor plant health and disease incidence in each treatment. Compare the effectiveness of foliar and drench applications in preventing or mitigating disease outbreaks.
REPLY TO COMMENT 4: We concur with the Reviewer's assertion that there is evidence supporting the notion that the utilization of various nanoparticles enhances plants' resistance to pests and diseases. Nevertheless, it is important to note that the current study did not take into account this particular aspect, and the treatments administered were not specifically intended to induce biotic stress on the tomato plants. As a result, the evaluation of the efficacy of TiO2 NPs in enhancing disease tolerance was not included in our analysis. It is deemed that this matter warrants further investigation.
5.- What’s the aims of maltodextrin (MDX) adding?
REPLY TO COMMENT 5: In the Introduction section, it was mentioned that a challenge encountered in the production of nanoparticles (NPs) of TiO2 is their tendency to agglomerate, hence diminishing their efficacy when utilized in plant applications. To provide further clarification on this topic, we have restated the goals in the following manner:
“The objective of this study was to investigate the impact of several methods of application, namely foliar and drench treatments, of nTiO2 on tomato plants. The study aimed to assess the impacts of these applications on plant development, nutritional status, gas exchange parameters, and fruit yield. In order to enhance the dispersibility and stability of nTiO2, a polysaccharide called maltodextrin (MDX) derived from maize starch was utilized to coat the surface of the NPs in order to mitigate the agglomeration of the NPs”
6.- How was measured SPAD chlorophyll index? It’s not clear
REPLY TO COMMENT 6: In accordance with the recommendation of the Reviewer, we have addressed this matter in subsection 4.7. The text has been revised and it is now:
“The SPAD index (SPAD-502 meter, Minolta, Japan), was recorded at 15-day intervals throughout the duration of the study. Two healthy, recently mature leaves were selected per experimental unit, and on each leaf, three randomly chosen spots were sampled to get the SPAD value”
7.- Material and method Lacks novelty in references
REPLY TO COMMENT 7: The Materials and Methods section contains a total of four references. The citations occurred in the years 1961, 1996, 1996, and 2007. The first citation may be traced back to a seminal study that has garnered over 1400 references since its initial publication. This work is deemed suitable for inclusion in the materials and methods section since it offers fundamental information pertaining to the formulation of nutrient solutions that are widely employed globally and that we used for the development of the nutrient solution utilized in the present investigation. Both articles published in 1996 align with the methodology commonly employed in laboratories for the mineral analysis of plant tissues, therefore, they are deemed essential to incorporate them in order to show the laboratory procedures utilized for the analysis of fruit tissues.
8.- The discussion lacks scientific explanations
REPLY TO COMMENT 8: It is essential to emphasize that the understanding of the impacts of nanoparticles on plants remains significantly constrained due to its status as an emerging technology, as previously mentioned in the introductory section; this implied that still many facts observed are not well understood. However, we consider that the discussion contains enough scientific explanations for the results we observed. For example, an attempt was made to elucidate the absence of treatment-induced effects on the concentration of titanium in the fruits by providing the following explanations: “Even though the Ti absorbed through the leaves and roots is reported to be translocated to other plant parts [11], the no effect on Ti concentration on the fruits observed when nTiO2 was applied via drench may be because it is preferentially accumulated in the roots, and only a small amount of Ti is transported to the shoot, as reported by Kelemen et al. [51].”. Another example is: “Another factor that may have played a role in the no effect on Ti concentration in the fruits is the particle size, as it has been demonstrated that using nTiO2 of large sizes may even cause a decrease in Ti content in leaves, stems and roots due to the large-diameter NPs are not absorbed by the root cells because they cannot fit the pore diameter of the cell wall, as demonstrated by Missaoui et al. [48] in fenugreek.”.
Reviewer 3 Report
The authors must seek the help of a scientific editor to help them produce a viable manuscript for publication. In addition, there is a need to provide a more nuanced introduction and justification of the study, with clear objectives. The materials and methods are sketchy and do not meet the normal standards of repeatability i.e., providing such detail and clarity that other researchers can repeat the study and validate the results of this study or otherwise. The presentation of the results and their discussion are similarly obtuse and require major revision
see the above comments
Author Response
1.- The authors must seek the help of a scientific editor to help them produce a viable manuscript for publication.
REPLY TO COMMENT 1: We acknowledge and value the proposals by the Reviewer, and we comprehend that, in his/hers perspective, our paper is not deemed suitable for publication. Nevertheless, we respectfully hold a divergent viewpoint and assert that the findings derived from our research possess significant value for publication. This is due to the fact that our work offers perspectives on the impact of nTiO2 on the physiological response of tomato plants. Numerous research pertaining to this subject have been published predominantly including short-term tests that fail to elucidate the impact on crucial factors such as fruit production, instead of focusing solely on vegetative characteristics. An additional advantage of our experiment is in the provision of empirical support for the notion that the coating of nTiO2 with maltodextrin boosts the responsiveness of plants; this enhancement is attributed to the reduction in the agglomeration tendency of these particular nanoparticles, which was achieved by the utilization of maltodextrin. Another significant achievement is that our data show that the nutrient status of plants treated with nTiO2 may appear not to be affected, however, when data are transformed to total nutrient content at least for some nutrients the effect of nTiO2 can bee observed.
2.- In addition, there is a need to provide a more nuanced introduction and justification of the study, with clear objectives.
REPLY TO COMMENT 2: The introduction was structured into four distinct paragraphs. The initial paragraph centers on delineating the present and forthcoming concerns pertaining to food production, and explores the potential of nanotechnology in offering viable solutions to address these issues. The second paragraph is centered on the utilization of nanotechnology as a significant means to address the prevailing challenges encountered in food production. The third paragraph centers on elucidating the efficacy of a particular nanomaterial, namely nTiO2, in augmenting the physiological and growth responses of various cultivated species. In the final paragraph, our attention is directed toward providing a comprehensive account of the agglomeration and stability issues that arise during the synthesis of nTiO2. This obstacle significantly reduces the effectiveness of this material. In the final paragraph, the study's objectives were outlined as follows: to demonstrate the potential reduction of nTiO2 agglomeration by the application of maltodextrin coating on nanoparticles, and to investigate the potential enhanced impact of this coated nTiO2 on plant development and physiology.
We believe that presenting the introduction in the manner that is outlined is the best way to inform the reader of the scope of the study. However, we concur with the Reviewer's assessment that the objectives were not sufficiently explicit and, as a result, they were subsequently modified. In the revised document, the objectives have been rephrased as follows: “The present study was designed to assess the effect of maltodextrin (MDX), a polysaccharide derived from maize starch, to form a coating on the surface of the nanoparticles in order to reduce the agglomeration of the nanoparticles and thus to improve the dispersibility and stability of nTiO2. Additionally, the study aimed to examine the effects of different application methods, specifically foliar and drench treatments, of nTiO2 on tomato plants, with the objective of evaluating the effects on plant growth, nutritional status, gas exchange parameters, and fruit production.”
In addition to this change, the introduction was revised to clarify the issues raised by the reviewer.
3.- The materials and methods are sketchy and do not meet the normal standards of repeatability i.e., providing such detail and clarity that other researchers can repeat the study and validate the results of this study or otherwise.
REPLY TO COMMENT 3: The Materials and Methods was revised and included the following information to provide more specific details about the procedures we conducted in the experiment. The details added were:
- The process of surface covering of the nTiO2 with maltodextrin was explained in detail
- The characterization process of the nanoparticles of nTiO2 with or without maltodextrin was explained in detail
- Information about substrate pH adjustment was incorporated
- Volume of nutrient solution used for irrigation on the vegetative and reproductive phases were incorporated
- Details about the frequency of irrigation were incorporated
- Acids used for pH adjustment of the nutrient solution were incorporated
- Brand of emitters used for irrigation was incorporated
- Details about leaching fraction calculation was incorporated
- Details about leaf pruning were incorporated
- Data logger specifications for monitoring environmental data were incorporated
- Specifications about the substrate components was incorporated
- Characteristics of the seedlings at transplant time were incorporated
- Information about the device for pH and EC measurements was incorporated
- Details about foliar sprays with n-TiO2 were incorporated
- Details about sampling of leaves and SPAD measurements were incorporated
- Details about processing the data from the 4 readings of gas exchange parameters with the IRGA were incorporated
- Timing about petiole sap sampling was incorporated
- Details about the treatments under study were incorporated
- Details about sampling and measurement of fruit firmness were incorporated
- Details about sampling and measurement of total soluble solids were incorporated
- Device used for measuring stem length was incorporated
- Device used for measuring stem diameter was incorporated
- Details about drying of fruits were incorporated
4.- The presentation of the results and their discussion are similarly obtuse and require major revision
REPLY TO COMMENT 4: The Results and Discussion sections underwent substantial revisions, including language editing and enhancements to improve overall coherence and fluency. A number of small errors in the Results section were rectified. In order to enhance clarity, modifications were made to two figures, while the tables now include additional facts that were previously absent in the earlier version. In order to enhance the coherence and readability of our paper, some measures were taken to streamline the Discussion part. This involved the removal of superfluous arguments in various subsections, as well as the relocation of certain paragraphs to more appropriate areas.
Round 2
Reviewer 2 Report
comments

Reviewer 3 Report
The author has done great work and the manuscript has been improved from the previous version. I am happy to accept the article.
No